# Poor Mobilizers in Lymphoma but Not Myeloma Patients Had Significantly Poorer Progression-Free Survival after Autologous Stem Cell Transplantation: Results of a Large Retrospective, Single-Center Observational Study

**DOI:** 10.3390/cancers15030608

**Published:** 2023-01-18

**Authors:** Normann Steiner, Georg Göbel, Leonie Mauser, Lena Mühlnikel, Marie Fischinger, Tina Künz, Wolfgang Willenbacher, Gabriele Hetzenauer, Jakob Rudzki, Walter Nussbaumer, Wolfgang Mayer, Eberhard Gunsilius, Brigitte Kircher, Dominik Wolf, David Nachbaur

**Affiliations:** 1Department of Internal Medicine V (Hematology and Medical Oncology), Medical University of Innsbruck, Anichstrasse 35, A-6020 Innsbruck, Austria; 2Department of Medical Statistics, Informatics and Health Economics, Medical University of Innsbruck, Schöpfstrasse 41/1, A-6020 Innsbruck, Austria; 3Central Institute for Blood Transfusion and Department of Immunology, University Hospital of Innsbruck, Anichstrasse 35, A-6020 Innsbruck, Austria

**Keywords:** lymphoma, multiple myeloma, autologous stem cell transplantation, granulocyte colony-stimulating factor, Plerixafor, poor mobilizer

## Abstract

**Simple Summary:**

Herein, we retrospectively analyze in our single-center study real-life data of 357 myeloma and lymphoma patients mobilized with granulocyte colony-stimulating factor plus a fixed dose of Plerixafor when indicated or G-CSF alone. There were no significant differences in engraftment kinetics or transfusion requirements between the Plerixafor Group and the G-CSF Group in the myeloma cohort. Lymphoma patients not requiring Plerixafor showed significantly faster neutrophil recovery, a trend for faster platelet recovery, and a significantly lower need for platelet transfusions. In myeloma patients, overall survival and progression-free survival after autologous stem cell transplantation were similar between the Plerixafor Group and the G-CSF Group, with hard to mobilize lymphoma patients showing significantly poorer progression-free survival and a trend also to lower overall survival.

**Abstract:**

In our single-center study, 357 myeloma and lymphoma patients between 2009 and 2019 were mobilized with granulocyte colony-stimulating factor (G-CSF 7.5 µg/kg bid for four days) plus a fixed dose of 24 mg Plerixafor when indicated (Plerixafor Group, *n* = 187) or G-CSF alone (G-CSF Group, *n* = 170). The target CD34 cell yields were ≥2.0 × 10^6^ CD34+ cells/kg in lymphoma and ≥4.0 × 10^6^ CD34+ cells/kg in myeloma patients to enable putative second transplants in the latter. There were no significant differences in engraftment kinetics or transfusion requirements between the Plerixafor Group and the control group in the myeloma cohort, with lymphoma patients not requiring Plerixafor showing significantly faster neutrophil recovery, a trend to faster platelet recovery, and a significantly lower need for platelet transfusions, probably due to the significantly lower number of CD34-positive cells re-transfused. While in myeloma patients the outcome (overall survival, progression-free survival) following autologous stem cell transplantation (ASCT) was similar between the Plerixafor Group and the control group, hard to mobilize lymphoma patients had significantly poorer progression-free survival (47% vs. 74% at 36 months after ASCT, *p* = 0.003) with a trend also to poorer overall survival (71% vs. 84%). In conclusion, while there seem to be no differences in stemness capacity and long-term engraftment efficiency between the Plerixafor and the G-CSF Group in lymphoma as well as myeloma patients, poor mobilizing lymphoma patients per se constitute a high-risk population with a poorer outcome after ASCT. Whether disease characteristics and/or a more intense or stem cell-toxic pre-mobilization chemo-/radiotherapy burden in this cohort are responsible for this observation remains to be shown in future studies.

## 1. Introduction

Multiple myeloma, a malignant hematologic disease, occurs mainly in the elderly and remains mostly incurable even with the availability of new drugs such as proteasome inhibitors (PI), immunomodulatory drugs (IMiDs), and monoclonal antibodies [1,2]. Front line autologous stem cell transplantation ASCT for transplant eligible myeloma patients up to 75 years of age following PI- and/or IMiD-based induction is still the treatment of choice resulting in continuously improved progression-free and overall survival (PFS, OS) [3,4,5].

Autologous stem cell transplantation is also the treatment of choice for chemo-sensitive relapses of diffuse large B cell lymphoma (DLBCL) with durable remissions of about 40% and for early and late relapses in transplant-eligible patients with Hodgkin’s disease (HD) [6,7,8,9]. In addition, young patients with advanced mantle cell lymphoma (MCL) and selected patients with follicular lymphoma (FL) and T-cell non-Hodgkin’s lymphoma (T-NHL) might be candidates for early ASCT or at the time of sensitive relapse [10,11,12]. 

Since 2009, Plerixafor^®^ (Mozobil) has been used in combination with G-CSF in poor mobilizers to improve the CD34+ cell yield [13]. Plerixafor, which is generally well tolerated, is an AMD300 bicyclic molecule and a selective and reversible CXCR4 antagonist and prevents its interaction with stroma-derived factor 1α, also known as CXCL12, resulting in an increased release of hematopoietic stem cells into the peripheral blood [14,15,16].

The aim of the present single-center study was to compare the outcome following ASCT in myeloma and lymphoma patients who needed Plerixafor for a successful stem cell mobilization procedure with the outcome in those myeloma and lymphoma patients who were successfully mobilized with G-CSF alone during the same time period. 

## 2. Patients and Methods

### 2.1. Patients

Between 2009 and 2019, 360 consecutive patients were included in this retrospective single-center analysis. Detailed patient characteristics are listed in Table 1 and Table 2. 

In the myeloma group, 108/211 (51%) patients and in the lymphoma group, 79/146 (54%) patients needed Plerixafor according to its labeled indication (additional administration of plerixafor was indicated in a CD34+ cell count ≤ 20 CD34+/μL in the peripheral blood (PB) on day 4 of mobilization with G-CSF alone or CD34+ cells remained ≤ 20 CD34+ cells/μL and no further increase was to be expected after chemo-mobilization despite administration of G-CSF for at least four days and a WBC ≥ 5.0 × 10 G/L, and if <1.0 × 10^6^/kg CD34+ cells were collected after the first apheresis) and clinical and published experience [17,18] in addition to G-CSF alone (steady state mobilization) or chemotherapy plus G-CSF to guarantee an optimal stem cell yield for one ASCT defined as ≥ 2.0 × 10^6^/kg CD34+ cells. The institutional standard dose for G-CSF was 7.5 µg administered subcutaneously bid and, when indicated, Plerixafor was used at a fixed dose of 24 mg administered subcutaneously late in the evening on day 4 to guarantee a delay of <11 h prior to the first or next leukapheresis. The conditioning regime for ASCT in myeloma patients was high-dose melphalan (100–200 mg/m²) and in lymphoma patients, the BEAM regime with all patients giving written informed consent. All myeloma patients but only 124/146 lymphoma patients proceeded to their first ASCT during the observation period (from 2009 until 2019). Neutrophil engraftment was defined as the first of two consecutive days with leukocytes ≥ 1.0 G/L and platelet engraftment was defined as the day of the last platelet transfusion. Approval for data collection and publication was obtained from the Ethics Committee of the Medical University of Innsbruck (vote # 1031/2020). 

### 2.2. Study Endpoints

The primary study endpoints were neutrophil and platelet engraftment kinetics and transfusion requirements after ASCT in both myeloma and lymphoma patients. The secondary endpoints contained OS, PFS, and secondary malignancies in the two cohorts.

### 2.3. Statistical Methods

We used descriptive statistics to analyze the samples and outcomes of the lymphoma and myeloma patients stratified for plerixafor use or not. All event summaries refer to the first sign of disease relapse/progression (PFS) or death (OS). To evaluate differences between the strata, we used appropriate non-parametric tests as well as univariate and multivariate survival models (Kaplan–Meier curves, Log-rank test, Cox regression). The Kolmogorov–Smirnov test was applied for testing of distributions for continuous variables. We defined OS as the time from day of transplant to day of death or date of last follow-up. We further defined PFS as the timespan from day of transplant to date of disease relapse/progression or last follow-up. Only patients who received an ASCT were included for calculation of PFS and OS. As described in [19], we estimated unadjusted cumulative 36-month risks for mortality and progression defined as the probability of the event within three years after transplantation. Additionally, we provide crude incidence rates as the number of events divided by the total number of person-years at risk after transplantation with 95% confidence intervals (CIs) according to a Poisson distribution. Using Cox proportional hazards models, we examined the hazard ratio (HR) associations between disease relapse/progression or death and type of intervention. The time scale for calculation of the Cox proportional hazards models was months from the day of transplant. The proportional hazards assumption was tested by inspecting Kaplan–Meier curves and using Schoenfeld residuals. All tests for statistical significance were two-sided. *p* values less than 0.05 were considered statistically significant, and for point estimators, we provide 95% CIs. Statistical evaluation was performed using SPSS version 27.0 statistical software (SPSS, Chicago, IL, USA) and Stata version 17 (StataCorp, College Station, TX, USA). 

## 3. Results

### 3.1. Efficacy of G-CSF +/− Plerixafor for Stem Cell Mobilization, Engraftment Kinetics, and Transfusion Requirements after ASCT

During the study period, 357/360 lymphoma and myeloma patients underwent a stem cell mobilization procedure at our institution. Three lymphoma patients receiving Plerixafor did not proceed to stem cell harvest and were therefore not included in the analysis. In the myeloma cohort, 108/211 (51%) patients required Plerixafor. The total median CD34+ cell number collected was 6.1 × 10^6^/kg (IQR, 4.8–8.2) with no significant difference between myeloma patients requiring Plerixafor or not with a success rate within a single apheresis procedure of 75% in the Plerixafor (CD34+ cell number collected was 6.5 × 10^6^/kg (IQR, 4.9–8.8)) and of 74% in the G-CSF Group (CD34+ cell number collected was 5.7 × 10^6^/kg (IQR, 4.8–7.7)) (Table 3). 

The overall success rate defined as ≥2 × 10^6^/kg CD34+ cells in patients with lymphoma was 87% in the Plerixafor Group compared to 100% in the G-CSF Group (*p* = 0.003, Table 4.). A single apheresis procedure was sufficient in 67% of patients in the Plerixafor Group compared to 91% of those in the G-CSF Group (*p* < 0.001) (Table 4). Overall, Plerixafor led to a 7-fold increase in CD34+ cell numbers in peripheral blood in the entire cohort (myeloma and lymphoma patients) with a significantly greater increase in myeloma than in lymphoma patients (8-fold vs. 4-fold, *p* < 0.001).

### 3.2. Autologous Stem Cell Transplantation and Engraftment Kinetics

During the study period, 92% of successfully mobilized patients proceeded to ASCT, namely 96% myeloma and 85% lymphoma patients. Thirty patients, all in the Plerixafor Group, did not proceed to transplant either because of patient refusal, transplant ineligibility, or for other reasons.

The median CD34+ cell number re-transfused per transplant in myeloma patients was similar in both groups, whereas in lymphoma patients a significantly lower CD34+ cell number was re-transfused in the Plerixafor Group (*p* = 0.03, Table 3 and Table 4).

Virtually all patients received either G-CSF (30 µg/d subcutaneously) from day +7 or in the case of age and comorbidities, pegfilgrastim 6 mg subcutaneously on day +1 to accelerate neutrophil recovery (Table 3 and Table 4).

While in myeloma patients there were no significant differences in engraftment kinetics or transfusion requirements between the Plerixafor Group and the G-CSF Group, lymphoma patients not requiring Plerixafor showed significantly faster neutrophil recovery, a trend to faster platelet recovery, and a significantly lower need for platelet transfusions (Table 3 and Table 4).

### 3.3. Survival Outcomes and Secondary Malignancies

The 3-year OS in myeloma patients was 84% in both the Plerixafor Group and the G-CSF Group (*p* = 0.9) (Figure 1).

The 3-year PFS in myeloma patients requiring Plerixafor was 58% (95% CI, 49–65%) vs. 46% (95% CI, 37–55%) in the G-CSF Group (*p* = 0.2) (Figure 1).

The 3-year OS in lymphoma patients was 71% (95% CI, 59–80%) in the Plerixafor Group and 84% (95% CI, 76–89%) in the G-CSF Group (*p* = 0.1) (Figure 2).

The 3-year PFS in lymphoma patients mobilized with Plerixafor was 47% (95% CI, 34–60%) vs. 74% (95% CI, 65–81%) in the G-CSF Group (*p* = 0.003) (Figure 2).

During the observation period, eight myeloma patients (4%, four in each cohort; Plerixafor Group: one melanoma, one prostate cancer, one lung adenocarcinoma, one renal cell carcinoma; G-CSF Group: one Hodgkin’s lymphoma, two melanomas, and one t-AML) and six (5%) lymphoma patients developed a secondary malignancy after ASCT (five in the Plerixafor Group: one non-small-cell lung carcinoma, one myelodysplastic syndrome, three other lymphomas, and one in the G-CSF Group with a myeloproliferative neoplasm) (Table 3 and Table 4).

## 4. Discussion

Patients with multiple myeloma, who are transplant-eligible, undergo ASCT in first line setting with the goal of achieving better PFS and OS [20].

In contrast, patients with aggressive lymphoma usually do not undergo ASCT in first line therapy but in selected cases in the relapse setting (second or later line) to improve the outcome [21,22,23]. Thus, it is of utmost importance that a sufficient CD34+ cell collection count be achieved in the early disease phase followed by cryopreservation, regardless of whether ASCT is planned soon or later.

The present single-center analysis demonstrates that the addition of Plerixafor to G-CSF in so-called ‘poor mobilizers’ according to its labeled indication permitted 92% of all myeloma and lymphoma patients requiring ASCT, either as part of their frontline treatment or as salvage treatment for chemo-sensitive relapse in order to prolong PFS and OS, were able to be successfully mobilized to guarantee prompt engraftment after transplant. This goal was achieved within a single leukapheresis in 75% of myeloma and in 67% of lymphoma patients. Lymphoma patients mobilized with Plerixafor significantly more frequently required a second apheresis than did myeloma patients. The reason is probably due to the more intensive pre-treatment with at least two lines of chemotherapy including aggressive salvage regimes. Our findings are in line with the report published by Hübel et al., who analyzed European data of poor stem cell mobilizers and confirmed a lesser collection success for non-Hodgkin’s lymphoma patients than for myeloma and Hodgkin’s lymphoma patients. They also demonstrated the effectiveness of Plerixafor in poorly mobilized patients to increase the pool of patients for whom an autologous stem cell transplantation is a valid therapy option [13]. 

In our study, median time to neutrophil engraftment was similar in both myeloma groups. Our findings are in line with those of Worel et al. with no significant difference in time to neutrophil engraftment between the mobilization cohorts [24]. Prakash et al. observed faster neutrophil engraftment in patients mobilized with Plerixafor than in those in the pre-Plerixafor control group [25]. However, it should be noted that the mentioned study compared its findings with a historical control group with no possibility for Plerixafor administration. The fact that in our study no delay in neutrophil engraftment in myeloma patients was observed can certainly be attributed to the sufficient bone marrow function in patients who were not intensively pre-treated. However, in lymphoma patients mobilized without Plerixafor neutrophil engraftment was significantly faster than in patients receiving Plerixafor. Our findings are in line with those of Yuan et al., who observed significantly faster neutrophil engraftment in lymphoma patients mobilized without Plerixafor [26]. Other studies have described similar neutrophil engraftment kinetics, but they observed no statistically significant differences between the groups [27,28,29,30,31,32]. Median time of platelet engraftment in our study did not significantly differ in either of the mobilized myeloma groups and is in line with that of other authors [24,25]. This can also be explained by the sufficient bone marrow function in patients who did not undergo intensive pre-treatment. 

Tricot et al. reported fast recovery of platelets within 14 days after high-dose cyclophosphamide and less than 12 months of prior chemotherapy as predictors of early engraftment [33]. 

On the other hand, lymphoma patients mobilized without Plerixafor showed a trend to faster engraftment. Results almost similar to our findings with a slight delay in the median time to platelet engraftment for the Plerixafor Group as compared to the control group were observed by Sureda and Yuan et al. [26,28]. Other reports showed similar platelet engraftment kinetics in the different mobilization lymphoma cohorts [29,30,31]. Importantly, in agreement with other studies, the cells collected after treatment with Plerixafor led to durable and fast neutrophil and platelet engraftment. Regarding the transfusion requirement, there were no significant differences in either of the mobilized myeloma groups. Our findings are in line with those of other authors with no significant differences in need of red cell or platelet transfusions [24,25].

However, a trend to higher transfusion requirements was observed in lymphoma patients mobilized with Plerixafor. The difference was not significant, admittedly, but could be explained as an expression of stem cell-toxic pretreatment in lymphoma patients. No significant differences in transfusion requirements in the different mobilization treatments were seen in other studies [24,26].

While there was no difference in OS or PFS in the myeloma cohort between patients requiring Plerixafor or not for successful stem cell mobilization, ‘hard to mobilize’ lymphoma patients requiring Plerixafor had a significantly poorer outcome regarding PFS and also a trend to poorer OS. In contrast to our data, Moreb et al. observed a significantly shortened PFS and OS in poorly as compared to well mobilized myeloma patients [34]. A reason for this difference can be the more aggressive disease biology, especially in poor mobilizers, including different risk factors for not sufficient mobilization with poorer outcome. Factors such as age >60, extensive prior treatment, thrombocytopenia, >1 line of induction treatment, prior radiation therapy, prior exposure to alkylating substances or melphalan and the prolonged use of lenalidomide are responsible for poor or suboptimal mobilization in myeloma patients [35,36,37,38,39] with the speculation that each risk factor results in cumulative effects for poor mobilization including poorer survival outcome. However, in the myeloma cohort, all these reflections argue against our results. Crocchiolo et al. observed a poorer outcome after allogeneic stem cell transplantation in myeloma and lymphoma patients with poor autologous mobilization status [40].

In our study, ‘hard to mobilize’ lymphoma patients showed a significantly poorer outcome for PFS and a trend to poorer OS. Our outcome findings are in line with those of other lymphoma studies, which showed in patients mobilized poorly with autologous stem cells a substantially shorter PFS and OS than in good mobilizers [28,32,41]. In our lymphoma cohort, the results can be explained by the advanced lymphoma disease and the larger number of patients included in the poor mobilizers cohort. Several risk factors for suboptimal or poor mobilization such as age over 60 years, progressive disease, bone marrow involvement, disease status and prior treatment, previous radiation, previous and type of chemotherapy, thrombocytopenia, neutropenic fever during the stem cell mobilization, failure of previous stem cell mobilization attempts, and patients not in remission after first line therapy are well known [13,41,42,43,44,45,46,47,48,49]. The presence of more than one of these factors including the lymphoma biology could be responsible for an unfavorable outcome. 

Regarding the incidence of secondary malignancies after Plerixafor and G-CSF therapy, only marginal data are available. During our study, only 4% of myeloma patients in both treatment cohorts developed a secondary malignancy. Lymphoma patients treated with Plerixafor might tend to develop a secondary malignancy (6% in the Plerixafor Group, 1.5% in the G-CSF Group), although there was no statistical significancy. Our findings are in line with those of Doel et al. who observed five patients with a secondary malignancy, reflecting a cumulative incidence of 17% [50]. As data on the development of secondary malignancies in patients undergoing stem cell transplantation after mobilization with Plerixafor are very sparse, they need to be investigated in controlled prospective studies.

In summary, the significantly poorer outcome in lymphoma patients requiring Plerixafor in addition to G-CSF for a sufficient stem cell mobilization procedure regarding PFS, the slower engraftment kinetics and the greater transfusion requirements might suppose that these patients probably had a significantly higher and more stem cell-toxic pre-mobilization chemo-/radiotherapy burden and probably per se had more aggressive lymphoma subtypes. The lack of late graft failures and the low incidence of secondary malignancies in both the Plerixafor and G-CSF subgroups suppose no obvious functional differences between Plerixafor + G-CSF- or G-CSF-mobilized long-term repopulating hematopoietic stem cells when used for ASCT. 

## 5. Conclusions

While there seem to be no differences in stemness capacity and long-term engraftment efficiency between the Plerixafor Group and the G-CSF Group in lymphoma as well as myeloma patients, poor mobilizing lymphoma patients per se constitute a high-risk population with a poorer outcome after ASCT.

## Figures and Tables

**Figure 1 cancers-15-00608-f001:**
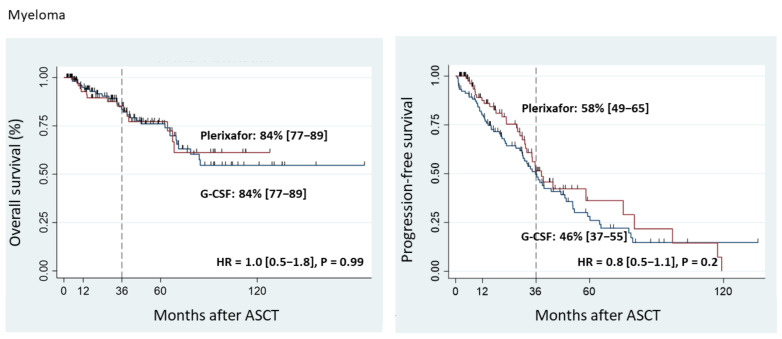
Overall survival and progression-free survival of myeloma patients.

**Figure 2 cancers-15-00608-f002:**
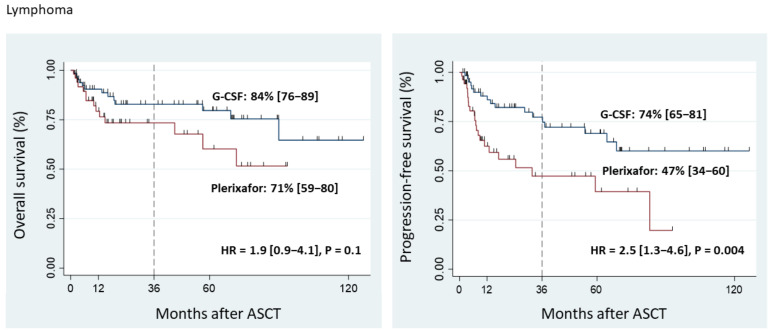
Overall survival and progression-free survival of lymphoma patients.

**Table 1 cancers-15-00608-t001:** Multiple Myeloma patients and disease characteristics.

Multiple Myeloma Patients	Plerixafor Group	G-CSF Group	*p* Value
**Number of patients—N (%)**	108 (51)	103 (49)	
**Age at diagnosis (years)—median [IQR]**	59 [52–64]	55 [47–62]	0.01
**Gender**			0.6
Male—N (%)	63 (58)	64 (62)	
Female—N (%)	45 (42)	39 (38)	
**Bone marrow infiltration at diagnosis**			0.4
Yes—N (%)	91 (85)	90 (88)	
No—N (%)	4 (4)	3 (3)	
n.a.	13 (12)	10 (10)	
**Disease stage at diagnosis (ISS)**			0.9
ISS I—N (%)	48 (44)	46 (45)	
ISS II—N (%)	34 (32)	28 (27)	
ISS III—N (%)	22 (20)	24 (23)	
n.a.—N (%)	4 (4)	5 (5)	
**Chemomobilization—N (%)**	0 (0)	0 (0)	
**Prior lines of therapy**			0.04
1—N (%)	108 (100)	97 (94)	
2—N (%)	0 (0)	3 (3)	
≥3—N (%)	0 (0)	3 (3)	
**Prior radiotherapy**			0.2
Yes—N (%)	39 (36)	28 (27)	
No—N (%)	69 (64)	75 (73)	
**Disease status at ASCT**			<0.001
CR, nCR, sCR—N (%)	42 (39)	16 (16)	
VGPR—N (%)	27 (25)	36 (35)	
PR—N (%)	28 (26)	26 (25)	
SD—N (%)	2 (2)	1 (1)	
PD—N (%)	8 (7)	5 (5)	
n.a.—N (%)	1 (1)	19 (18)	

nCR, near complete remission; sCR, stringent complete remission; CR, complete remission; VGPR, very good partial remission; PR, partial remission; SD, stable disease; PD, progressive disease; n.a., not available; G-CSF, granulocyte colony-stimulating factor.

**Table 2 cancers-15-00608-t002:** Lymphoma patients and disease characteristics.

Lymphoma Patients	Plerixafor Group	G-CSF Group	*p* Value
**Number of Patients** —N (%)	82 (55)	67 (45)	
DLBCL	29 (35)	17 (25)	
MCL	10 (12)	15 (22)	
FL	8 (10)	5 (8)	
HL	9 (11)	4 (6)	
Burkitt’s lymphoma	5 (6)	6 (9)	
AITL	5 (6)	3 (5)	
TLBL	7 (9)	0 (0)	
PTCL	4 (5)	2 (3)	
Primary CNS lymphoma	1 (1)	4 (6)	
Other B/T-NHL	4 (5)	11 (16)	
**Age at diagnosis (years)—median [IQR]**	52 [44–58]	49 [41–57]	0.4
**Gender**			0.8
Male—N (%)	57 (70)	45 (67)	
Female—N (%)	25 (30)	22 (33)	
**Bone marrow infiltration at diagnosis**			0.4
Yes—N (%)	12 (15)	16 (24)	
No—N (%)	48 (58)	35 (52)	
n.a.	22 (27)	16 (24)	
**Disease stage at diagnosis** (Ann Arbor)			0.4
I–II N (%)	18 (22)	7 (11)	
III–IV N (%)	60 (74)	42 (62)	
n.a.—N (%)	4 (5)	18 (27)	
**Chemomobilization**—N (%)	82 (100)	67 (100)	
**Prior lines of therapy**			0.8
1—N (%)	22 (27)	16 (24)	
2—N (%)	43 (52)	39 (58)	
≥3—N (%)	17 (21)	12 (18)	
**Prior radiotherapy**			0.8
Yes—N (%)	11 (13)	10 (15)	
No—N (%)	71 (87)	57 (85)	
**Disease status at ASCT**			0.08
CR—N (%)	40 (49)	29 (43)	
PR—N (%)	25 (30.5)	21 (31)	
SD—N (%)	8 (10)	1 (1.5)	
Mixed response—N (%)	0 (0)	1 (1.5)	
PD—N (%)	2 (2)	5 (8)	

CR, complete remission; PR, partial remission; SD, stable disease; PD, progressive disease; n.a., not available; DLBCL, diffuse large B-Cell lymphoma; MCL, mantle cell lymphoma; FL, follicular lymphoma; HL, Hodgkin’s lymphoma; AITL, angioimmunoblastic T-Cell lymphoma; TLBL, T-Cell lymphoblastic lymphoma; PTCL, peripheral T-Cell lymphoma; NHL, non-Hodgkin’s lymphoma; G-CSF, granulocyte colony-stimulating factor.

**Table 3 cancers-15-00608-t003:** CD34+ cell kinetics, engraftment, and outcome in multiple myeloma patients.

Multiple Myeloma Patients	Plerixafor Group	G-CSF Group	*p* Value
**Number of stem cell mobilized patients—N (%)**	108 (51)	103 (49)	
**Total number of CD34+ cells collected (×106/kg)—median [IQR]**	6.5 [4.9–8.8]	5.7 [4.8–7.7]	0.2
**Total number of apheresis procedures**			0.9
1—N (%)	83 (77)	78 (76)	
2—N (%)	19 (17.5)	20 (19)	
3—N (%)	6 (5.5)	5 (5)	
**Success defined as:**			
≥4 × 106/kg CD34+ cells—N (%)	93 (86)	98 (95)	0.03
Success in a single apheresis procedure—N (%)	81 (75)	76 (74)	0.8
**Number of patients receiving a first ASCT**	100 (93)	103 (100)	0.02
**Transplanted CD34+ cell number (×106/kg)—median [IQR]**	3.5 [2.7–4.9]	3.8 [2.6–5.3]	0.9
**Time to neutrophil engraftment—median [range]**	12 [8–15]	12 [9–20]	0.9
**Time to platelet engraftment—median [range]**	12 [8–25]	11 [8–34]	0.1
**Number of red cell transfusions—median [range]**	1.5 [0–12]	0 [0–10]	0.6
**Number of platelet transfusions—median [range]**	2 [0–14]	2 [0–12]	0.1
**3-year progression-free survival—months % (95% CI)**	58 (49–65)	46 (37–55)	0.2
**3-year overall survival—months % (95% CI)**	84 (77–89)	84 (77–89)	0.9
**Secondary malignancies**—N (%)	4 (4)	4(4)	0.9

ASCT, autologous stem cell transplantation; G-CSF, granulocyte colony-stimulating factor. In lymphoma patients, the median total CD34+ cell number collected was 4.4 × 10^6^/kg (IQR, 2.5–7.8) with a significantly lower CD34+ cell number harvested in patients requiring Plerixafor (3.3 vs. 5.6, *p* < 0.001) (Table 4).

**Table 4 cancers-15-00608-t004:** CD34+ cell kinetics, engraftment, and outcome in lymphoma patients.

Lymphoma Patients	Plerixafor Group	G-CSF Group	*p* Value
**Number of stem cell mobilized patients—N (%)**	79 (54)	67 (46)	
**Total number of CD34+ cells collected (×106/kg)—median [IQR]**	3.3 [2.2–6.1]	5.6 [3.4–11.0]	<0.001
**Total number of apheresis procedures**			0.02
1—N (%)	52 (66)	58 (87)	
2—N (%)	24 (30)	7 (10)	
3—N (%)	2 (3)	2 (3)	
4—N (%)	1 (1)	0 (0)	
**Success defined as:**			
≥2 × 106/kg CD34+ cells—N (%)	69 (87)	67 (100)	0.003
Success in a single apheresis procedure—N (%)	52 (67)	60 (91)	<0.001
**Number of patients receiving a first ASCT—N (%)**	57 (79)	67 (100)	<0.001
**Transplanted CD34+ cell number (×106/kg)—median [IQR]**	4.0 [2.4–6.2]	5.2 [3.2–9.3]	0.03
**Time to neutrophil engraftment—median [range]**	11 [8–14]	10 [8–16]	0.0004
**Time to platelet engraftment—median [range]**	13 [5–59]	12 [5–17]	0.04
**Number of red cell transfusions—median [range]**	4 [0–24]	2 [0–10]	0.43
**Number of platelet transfusions—median [range]**	5 [1–54]	3 [1–20]	0.01
**3-year progression-free survival—months % (95% CI)**	47 (34–60)	74 (65–81)	0.003
**3-year overall survival—months % (95% CI)**	71 (59–80)	84 (76–89)	0.1
**Secondary malignancies**	5 (6)	1 (1.5)	0.1

ASCT, autologous stem cell transplantation; G-CSF, granulocyte colony-stimulating factor.

## Data Availability

The generated or analyzed data are available from the corresponding author on reasonable request.

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
