# Peer review of "Poor Mobilizers in Lymphoma but Not Myeloma Patients Had Significantly Poorer Progression-Free Survival after Autologous Stem Cell Transplantation: Results of a Large Retrospective, Single-Center Observational Study"

_cancers, 2023, doi:10.3390/cancers15030608_

Round 1

Reviewer 1 Report

To authors,

This is a single institute retrospective study to compare the survivals between multiple myeloma or lymphoma patients who needed to use plerixafor for stem cell collection and who did not. Although this is a retrospective study, the patients’ numbers were relatively high so that this study may show useful information if it was analyzed correctly. However, there are several concerns for this paper to publish in this way.

Major points

1)    The main message of this paper is that “hard to mobilize” lymphoma patients showed poorer outcomes. However, as the authors say, the reasons might be underlying disease status and not usage of plerixafor.  

2)    Twenty-two out of 79 patients in plerixafor group did not proceed to ASCT. This must directly impact on the outcome.

3)    The lymphoma patients in plerixafor group showed higher incidence of secondary malignancy (5 out of 79=6%). However, again, 22 out of 79 patients in plerixafor group who did not proceed to ASCT were included in this analysis. ASCT itself should have impacts on the incidence of secondary malignancy. This seems inappropriate to compare the incidence of secondary malignancy in this way.

Minor points

1)    Page 1 Line 29: G-CSF 7,5μg should be G-CSF 7.5μg

2)    In the tables, Filgastrim should be Filgrastim

3)    Control group or Filgrastim group or G-CSF group should be called as the same name.

4)    In the title, it says it is a single-center study. It should be written in a main text too.

5)    In table 1 and 2, could you provide the disease status at apheresis?

6)    Page 5 Line 88: Plerixafor labelled indication should be written in the text.

7)    Page 5 Line 93: What is “based on the centers’ experience”? Could you define that?

8)    Page 5 Line 102: The observation period was not defined in method section.

9)    Page 5 Line 109-110: The transfusion requirements are in both primary and secondary endpoint.

10) Table 3 the second last row: “3 year – Overall survival of 36 months” should be “3 year-Overall survival”?

11) Table 4 after third row, “total number of apheresis procedures” was missing.

12) Page 7 Line 155-157. I do not understand from where these results came.

13) Page 9 Line 193. There should be a reference.   

Author Response

Poor mobilizers in lymphoma but not myeloma patients had significantly poorer progression-free survival after autologous stem cell transplantation: results of a large retrospective, single-center observational study

Reviewer #1

Response to reviewer comments

This is a single institute retrospective study to compare the survivals between multiple myeloma or lymphoma patients who needed to use plerixafor for stem cell collection and who did not. Although this is a retrospective study, the patients’ numbers were relatively high so that this study may show useful information if it was analyzed correctly. However, there are several concerns for this paper to publish in this way.

We thank the Reviewer for the positive feedback and the opportunity to optimize our paper.

Major points

  • The main message of this paper is that “hard to mobilize” lymphoma patients showed poorer outcomes. However, as the authors say, the reasons might be underlying disease status and not usage of plerixafor.  

We thank the Reviewer for this important remark and totally agree that “hard to mobilize” lymphoma patients have a poorer outcome. Of course, this is not related to the use of Plerixafor, as we mentioned in our paper. This remark has now been clearly elaborated and the somewhat unfortunate title has been changed for the sake of clarity.

  • Twenty-two out of 79 patients in plerixafor group did not proceed to ASCT. This must directly impact on the outcome.

We thank the Reviewer for this critical comment. In our study, 22 of 79 patients did not undergo ASCT due to insufficient stem cell yield or disease progression. These 22 patients were not included for calculation of PFS and OS. Exclusion of the 22 patients had no direct impact on the outcome of lymphoma patients, as PFS and OS were calculated starting from day 0 of stem cell transplantation.

We now clearly describe this procedure in the Methods section of the paper.

  • The lymphoma patients in plerixafor group showed higher incidence of secondary malignancy (5 out of 79=6%). However, again, 22 out of 79 patients in plerixafor group who did not proceed to ASCT were included in this analysis. ASCT itself should have impacts on the incidence of secondary malignancy. This seems inappropriate to compare the incidence of secondary malignancy in this way.

We thank the Reviewer for this critical comment.

As mentioned in the previous point, the 22 patients were not included in the analysis of PFS and OS and also not in calculation of the incidence of secondary malignancy. Thus, a comparison of the incidence of secondary malignancy is appropriate in both groups of patients who underwent ASCT.

If the 22 patients who did not undergo ASCT had been included in calculation of the incidence of secondary malignancy, a comparison of the two groups would, of course, as the Reviewer correctly pointed out, be inappropriate. We have clearly stated this point in the paper.

Minor points

  • Page 1 Line 29: G-CSF 7,5μg should be G-CSF 7.5μg

Thank you for this remark. The correction has been made.

  • In the tables, Filgastrim should be Filgrastim

Thank you for this remark. The wording is now consistent throughout the paper: G-CSF Group.

  • Control group or Filgrastim group or G-CSF group should be called as the same name.

We apologize for the confusion concerning the “different groups.” The wording is now consistent throughout the paper: G-CSF Group.

  • In the title, it says it is a single-center study. It should be written in a main text too.

Thank you. This suggestion has now been incorporated in the paper.

  • In table 1 and 2, could you provide the disease status at apheresis?

Since this was a retrospective study, the disease status at apheresis was not routinely collected at our center. Instead, we collected the disease status at the time of transplantation and this is shown in the tables.

  • Page 5 Line 88: Plerixafor labelled indication should be written in the text.

The Plerixafor labelled indication is now clearly stated in the paper.

  • Page 5 Line 93: What is “based on the centers’ experience”? Could you define that?

Thank you for this remark. We have now deleted this inappropriately worded sentence from the paper.

  • Page 5 Line 102: The observation period was not defined in method section.

The observation period is now clearly stated in the paper.

  • Page 5 Line 109-110: The transfusion requirements are in both primary and secondary endpoint.

The transfusion requirements are now mentioned as secondary endpoint.

  • Table 3 the second last row: “3 year – Overall survival of 36 months” should be “3 year-Overall survival”?

Thank you for this remark. The correction has now been made.

  • Table 4 after third row, “total number of apheresis procedures” was missing.

Thank you for this remark. The correction has now been made.

  • Page 7 Line 155-157. I do not understand from where these results came.

These results are now clearly stated in the paper.

  • Page 9 Line 193. There should be a reference.   

A reference is now included.

We would greatly appreciate your considering our revised work for publication.

Reviewer 2 Report

The authors provided a detailed real-world retrospective analysis of engraftment kinetics and transfusion requirements but also outcomes after autologous transplantation in patients undergoing stem-cell mobilisation with or without plerixafor in a single-institution. 

The analysis included 357 patients, 187 of which were stimulated with a fix dose of 24 mg plerixafor. The authors noticed no differences in engraftment kinetics or transfusion requirements. The lymphoma patients from the plerixafor group showed lower yields of CD34+ cells/kg body weight, poorer outcomes in terms of PFS, longer time to neutrophil engraftment and required more platelet transfusions.

Minor issues: 

line 59: cure rates should be rephrased to e.g. durable remissions

Table 1: there is a significant age difference at diagnosis of myeloma patients in the plerixafor and in the control group - could you comment on this?

lines 138 -144: it is not clear if the reported data are describing the whole cohort (myeloma+ lymphoma) or myeloma. I assume that the data are referring to myeloma. 

line 142-143: for consistency reasons I recommend the authors to include the collected CD34+ counts for plerixafor and control group.

Table 4: the caption title: Total number of apheresis procedures is missing 

Line 152: please define the term “overall success rate”

Lines 154-157: Plerixafor led to a 7-fold increase in the CD34+ cell numbers: does this statement refer to the CD34+ cells in the peripheral blood or in the yields? 

Line 166: all patients received either G-CSF (30µg/d subcutaneously)… Please check, if this dosage was appropriate for overweight patients. Furthermore, in the line 96 the abbrevitation s.c. was used, in further text “subcutaneously”. 

Lines 187-189: the authors report on total 14 patients with secondary malignancies. I recommend to list the diagnoses of secondary malignancies. 

Lines 191-193. “Patients with hematologic diseases, whether myeloma or lymphoma, who are eligible for ASCT, undergo either in first line or in the relapsed setting (second or later line) ASCT with the goal….” The patients with aggressive lymphoma usually do not undergo ASCT in first therapy line. Please rephrase this sentence. 

Line 224: I think there is a typo: “significant differences between there (three?)”

Other issues: please report how many patients underwent second mobilisation attempt - not the collection following day (e.g. after frustraneous chemomobilisation)? Were there any grafts consisting from both G-CSF and plerixafor mobilization (i.e. G-CSF collection one day and plerixafor mobilization the day after?)

Author Response

Poor mobilizers in lymphoma but not myeloma patients had significantly poorer progression-free survival after autologous stem cell transplantation: results of a large retrospective, single-center observational study

Reviewer #2

Response to reviewer comments

The authors provided a detailed real-world retrospective analysis of engraftment kinetics and transfusion requirements but also outcomes after autologous transplantation in patients undergoing stem-cell mobilisation with or without plerixafor in a single-institution. 

The analysis included 357 patients, 187 of which were stimulated with a fix dose of 24 mg plerixafor. The authors noticed no differences in engraftment kinetics or transfusion requirements. The lymphoma patients from the plerixafor group showed lower yields of CD34+ cells/kg body weight, poorer outcomes in terms of PFS, longer time to neutrophil engraftment and required more platelet transfusions.

We thank the Reviewer for the positive feedback and the opportunity to optimize our paper.

Minor issues: 

line 59: cure rates should be rephrased to e.g. durable remissions

Thank you for this remark. The correction has now been made.

Table 1: there is a significant age difference at diagnosis of myeloma patients in the plerixafor and in the control group - could you comment on this?

The significant age difference at time of diagnosis is related to the retrospective observational design of the study. As there was no difference in treatment strategy related to age in this cohort, the difference can be explained as random.

lines 138 -144: it is not clear if the reported data are describing the whole cohort (myeloma+ lymphoma) or myeloma. I assume that the data are referring to myeloma. 

This point is now clearly stated in the paper.

line 142-143: for consistency reasons I recommend the authors to include the collected CD34+ counts for plerixafor and control group.

Thank you for this remark. The suggestion has now been incorporated in the paper.

Table 4: the caption title: Total number of apheresis procedures is missing 

Thank you for this remark. The correction has now been made.

Line 152: please define the term “overall success rate”

The “overall success rate” is now defined in the paper.

Lines 154-157: Plerixafor led to a 7-fold increase in the CD34+ cell numbers: does this statement refer to the CD34+ cells in the peripheral blood or in the yields? 

Thank you for this comment. The 7-fold increase in the CD34+ cell numbers refers to the CD34+ cells in the peripheral blood. We now clearly state this in the paper.

Line 166: all patients received either G-CSF (30µg/d subcutaneously)… Please check, if this dosage was appropriate for overweight patients. Furthermore, in the line 96 the abbrevitation s.c. was used, in further text “subcutaneously”. 

Since data collection was retrospective and complete data documentation was unfortunately not given, the weight of the patients, especially for overweight patients at the time of GCSF administration for apheresis, cannot be tracked retrospectively.

We thank you for pointing out that s.c. was not worded uniformly. We have now changed this in the paper.

Lines 187-189: the authors report on total 14 patients with secondary malignancies. I recommend to list the diagnoses of secondary malignancies. 

Thank you for this remark. The secondary malignancies are now listed in the paper.

Lines 191-193. “Patients with hematologic diseases, whether myeloma or lymphoma, who are eligible for ASCT, undergo either in first line or in the relapsed setting (second or later line) ASCT with the goal….” The patients with aggressive lymphoma usually do not undergo ASCT in first therapy line. Please rephrase this sentence

Thank you for this remark. We have now rephrased the above-mentioned sentence.

Line 224: I think there is a typo: “significant differences between there (three?)”

Thank you for this remark. The correction has now been made. Here you find the correct sentence: “Other studies have described similar neutrophil engraftment kinetics, but they observed no statistically significant differences between the groups”.

Other issues: please report how many patients underwent second mobilisation attempt - not the collection following day (e.g. after frustraneous chemomobilisation)? Were there any grafts consisting from both G-CSF and plerixafor mobilization (i.e. G-CSF collection one day and plerixafor mobilization the day after?)

None of the patients required a second mobilization procedure, but many patients required more than one apheresis to obtain the minimal amount of stem cells for one (lymphoma patients) or two ASCTs (myeloma patients) whenever possible (see Tables 3 and 4).

We would greatly appreciate your considering our revised work for publication.

Round 2

Reviewer 1 Report

The authors have addressed all my concerns and questions. There are new small errors which should be corrected easily. 

P1 L31 2.0x106 should be 2.0x106

P1 L32 4.0x106 should be 4.0x106